# Study on the Quality of Mixed Silage of Rapeseed with Alfalfa or Myriophyllum

**DOI:** 10.3390/ijerph20053884

**Published:** 2023-02-22

**Authors:** Siwei Chen, Chen Wan, Yingjun Ma, Keqiang Zhang, Feng Wang, Shizhou Shen

**Affiliations:** 1Agro-Environmental Protection Institute, Ministry of Agriculture and Rural Affairs, Tianjin 300191, China; 2College of Resources and Environment, Yunnan Agricultural University, Kunming 650201, China; 3Dali Yunnan, Agricultural Ecosystem, National Observation and Research Station, Dali 671004, China; 4College of Resources and Environment, Northeast Agricultural University, Harbin 150030, China

**Keywords:** rapeseed, alfalfa, myriophyllum spicatum, mixed silage

## Abstract

The objective of this study was to improve the comprehensive rate of utilization of rapeseed (*Brassica napus subsp. napus* L.), Myriophyllum (*Myriophyllum spicatum* L.) spicatum and alfalfa (*Medicago sativa* L.), reduce resource waste and environmental pollution. In this experiment, the effects of different proportions of the mixed silage of rapeseed and alfalfa or M. spicatum on the fermentation and nutritional quality were analyzed and further improved the quality of mixed silage using molasses and urea. Rapeseed was separately silaged with alfalfa and M. spicatum based on the ratios of 3:7, 5:5 and 7:3. After 60 days of mixed silage, the fermentation index and nutrient contents were measured to explore the appropriate ratio of mixed silage. The mixing ratio of rapeseed and alfalfa was better at 3:7: The contents of NH_3_-N/TN (4.61%), lactic acid (96.46 g·kg^−1^ dry matter [DM]) were significantly higher (*p* < 0.05). The crude protein content (118.20 g·kg^−1^ DM) was the highest (*p* < 0.05), while the pH (4.56) was the lowest when the mixing ratio of rapeseed and M. spicatum was 7:3. Considering the fermentation and nutrition quality, it is suggested that rapeseed and alfalfa should be mixed as silage at a ratio of 3:7 with 3% molasses and 0.3% urea, and rapeseed and M. spicatum should be mixed as silage at a ratio of 7:3 with 3% molasses.

## 1. Introduction

China produces nearly 5 billion tons of agricultural waste each year [1]. The problem of agricultural pollution caused by the irrational use of agricultural wastes has become increasingly prominent. It not only causes serious water, soil and air pollution, but also poses a threat to human health and the sustainable development of agriculture [2]. However, as an important biomass resource, the utilization of agricultural waste has enormous potential and value [3]. 

Mixed silage, as a common agricultural waste feed technology, can comprehensively utilize agricultural wastes and obtain high quality silage [4]. The optimization of silage technology can enhance resource efficiency and feed quality, which has recently been widely researched. Molasses and urea are commonly used as nutritional additives in silage. For some feeds with low carbohydrate content, molasses is often added to effectively compensate for the lack of silage fermentation substrate and improve the silage quality [5], while feeds with low protein content often use urea to provide a nitrogen source for the feed and improve its nutritional value [6].

Rapeseed and myriophyllum spicatum are typical agricultural wastes with high nutritional value, while lacking in studies on their utilization. 

As the largest oil crop in the world, rapeseed (*Brassica napus subsp. napus* L.) has a large planting area and wide distribution throughout the world. The European Economic Community (EEC) reported that the global production of rapeseed was expected to be as high as 68 million tons from 2021 to 2022, with China accounting for 19.34%. Extracting edible oil after cutting and harvesting seeds is the most traditional mode of utilizing rapeseed [7], while the rate of utilization of the rapeseed straw is relatively low (<10%), and the straw is often discarded or directly burned, which causes substantial resource waste and environmental pollution [8,9]. A substantial amount of research has shown that the whole rapeseed plant is a worthwhile source of roughage. The whole plant is a forage resource worth developing [10], and its straw contains 5.24% crude protein (CP), which is higher than those of corn (*Zea mays* L.) straw (5%) and wheat (*Triticum aestivum* L.) straw (3.6%) [11]. Current research on rapeseed is primarily focused on cultivation techniques and compositional analysis, with less research on the use of rape as fodder.

Myriophyllum (*Myriophyllum spicatum* L.) is a perennial submerged herb, which grows so quickly and reproduces very effectively [12] that it threatens the original animals and plants. Therefore, most myriophyllum spicatum have often been discarded directly after harvesting to restrain their wanton growth, which causes a serious waste of agricultural resources. However, Koyama and Gallegos found that these plants have a high content of CP, and they also contain a variety of amino acids and trace elements, which can be exploited as a functional product for use as feed [12,13]. However, there have been few studies on the use of M. spicatum in feed preparation at this stage, and ample research is needed on this aspect of their use. 

In addition to the above two plants, the utilization potential of alfalfa is also worth attention. Alfalfa (*Medicago sativa* L.) is known as the “king of pasture” because it is rich in vitamins, minerals, protein and other nutrients, and it also has high yields, good palatability and high digestibility (70~80%) [14]. In recent years, the urgent need for the development of the dairy industry has accelerated the promotion and planting of alfalfa [15]. The traditional use of alfalfa is to make hay, but climate factors have led to the loss of nutrients in alfalfa during the process of drying, which leads to lower feed value and a waste of resources [16]. Silage is less affected by climate, but it is not easy to silage alfalfa alone because of its high protein content, high buffer energy value and low content of soluble carbohydrate [17]. Research has shown that alfalfa can improve the success rate of silage and nutritional value through mixed silage technology [15,18].

This experiment investigated the effects of mixing different ratios of rapeseed with alfalfa and *M. spicatum* on their nutrition and fermentation quality, and the appropriate parameters for the proportions of mixed silage were screened out, respectively. Simultaneously, molasses and urea were further used to regulate the quality of the two groups of mixed silage under their optimal proportions. 

## 2. Materials and Methods

### 2.1. Silage Material Treatment

Rapeseed (variety: Zhongyou 19) and alfalfa (variety: Aurora) were planted in the experimental field of the National Field Scientific Observation and Research Station of Agricultural Ecosystem in Dali, Yunnan Province, China (100°7′50′ E°49′47′ N°25°49′47′ N). The rapeseed plants were harvested after the seeds had been collected, while the alfalfa was harvested during its full flowering period. *M. spicatum* grows in ponds near the Erhai River basin and was cut during the mature period. After cutting, the three silage materials were naturally air-dried (1~2 days) to an appropriate moisture content and then cut into pieces (1~2 cm) with a hay cutter before silage. The silage was prepared on 3 April 2022, and the primary chemical components of the material were determined before silage. The chemical composition of the raw materials is shown in Table 1.

### 2.2. Experimental Design

Test 1: Three control groups were established, and 500 g of chopped rapeseed, alfalfa and *M. spicatum* were placed in polyethylene cans (diameter × cover height: 95 mm × 100 mm) for silage. The crops were designated SY for rapeseed, SX for alfalfa and SZ for *M. spicatum.* Six experimental groups were established. Chopped rapeseed and alfalfa were evenly mixed (fresh weight) at a ratio of 3:7 (M1), 5:5 (M2) and 7:3 (M3); mix chopped rapeseed and *M. spicatum* at a ratio of 3:7 (M4), 5:5 (M5) and 7:3 (M6) (fresh weight). The total weight of each treatment was 500 g, which was fully and evenly mixed, and then packed into polyethylene green storage tanks for compaction (density ≥ 500 kg/m^3^), covered inside and outside, sealed, stored in darkness, and fermented at a constant temperature room for 60 days. There were nine treatments in total, and each treatment was conducted in triplicate. The experimental design is shown in Table 2.

Test 2: In test 1, the mixed silage groups of rapeseed and alfalfa, rapeseed, and *M. spicatum* had relatively good fermentation quality, at ratios of 3:7 and 7:3, respectively (data support validation in this paper). The second experiment was conducted to improve the quality of mixed silage based on the results of the first experiment. The treatments included a rapeseed and alfalfa (3:7) test group, a control group (C1), a 0.3% urea treatment group (U1), a 3% molasses treatment group (M1) and a 0.3% urea + 3% molasses mixed treatment group (UM1). The rapeseed and *M. spicatum* (7:3) test group included the control group (C2), 0.3% urea treatment group (U2), 3% molasses treatment group (M2) and the 0.3% urea + 3% molasses mixed treatment group (UM2). There were 9 treatments in total, and each treatment was conducted in triplicate. The experimental design is shown in Table 3.

### 2.3. Measurement Index and Method

The sensory evaluation of silage was conducted after 60 days of fermentation using the sensory evaluation method developed by the German National Agricultural Association that used humans to evaluate the smell, structure and color of the silage [19].

One fermentation sample was utilized to determine the primary nutritional indicators. The DM content was measured by oven drying and dried to a constant weight at 65 °C for 48 h; the water soluble carbohydrate (WSC) content was determined by the anthrone-sulfuric acid method [20]. The contents of neutral detergent fiber (NDF) and acid detergent fiber (ADF) were determined as described by Van Soest [21]. The content of total nitrogen (TN) was determined by the Kjeldahl method [22], and the content of CP = TN × 6.25.

Another fermentation sample was added with distilled water for mixing and pulping based on a material-liquid ratio of 1:9 (mass-volume ratio), then centrifuged for 10 min at 10,000 rpm, and then passed through four layers of gauze. The supernatant was filtered to obtain the fermentation extract, which was used to determine the fermentation indices. The pH of the extraction solution was determined with an acidimeter. The contents of lactic acid (LA), acetic acid (AA), propionic acid (PA) and butyric acid (BA) were determined by high-performance liquid chromatography (HPLC) with an injection volume of 10 μL, a column temperature of 0 °C and a wavelength of 210 nm. The mobile phase was 0.2% phosphoric acid with a flow rate of 1 mL·min^−1^ [23]. The content of ammonia nitrogen (NH_3_-N) was determined as described by Broderick [24]. To evaluate the quality of mixed silage after 60 days, this experiment scored the fermentation quality of mixed silage using the V-Score scoring system formulated by the Japan Grassland Livestock Association [25]. Simultaneously, the fuzzy membership function method was used to comprehensively evaluate the nutrition and fermentation quality of silage [26].

### 2.4. Statistical Analysis

All the data were presented as an average of the replicate tests and analyzed by one-way and two-way analyses of variance (ANOVA) using SPSS 20.0 (IBM, Inc., Armonk, NY, USA); Origin2021 (OriginLab, Northampton, MA, USA) was used for drawing. In addition, the least significant difference (LSD) tests were at *p* < 0.05 using the SAS program version 9.1 (SAS Institute, Cary, NC, USA). All the symbols mentioned in this study and their description are presented in Table 4. 

## 3. Results

### 3.1. Chemical Composition of the Raw Materials for Sileage

The chemical composition of the three raw materials for silage is shown in Table 1. Rapeseed had the lowest contents of DM and CP(*p* < 0.05), which were 274.03 g·kg^−1^ FM and 51.15 g·kg^−1^ DM. The contents of DM, CP and WSC in alfalfa were significantly higher than the other treatments (*p* < 0.05), with CP reaching 551.63 g·kg^−1^. Compared with that in alfalfa and rapeseed, the content of WSC (149.70 g·kg^−1^ DM) in M. spicatum was lower, but the contents of DM and CP of M. spicatum were significantly higher than those of rapeseed (*p* < 0.05). The contents of ADF and DNF of M. spicatum were significantly lower than those of rapeseed and alfalfa (*p* < 0.05).

### 3.2. Sensory Evaluation of Silage

Samples were taken after 60 days of silage fermentation at different mixing ratios of rapeseed and alfalfa or *M. spicatum*, and the sensory quality of different treatments varied (Figure 1). The sensory score and rating results are shown in Table 5.

### 3.3. Analysis of the Nutritional Quality of the Mixed Silage

As shown in Figure 2a, the content of DM in SY and SZ were significantly lower at the end of silage (272.97 and 263.50 g·kg^−1^ FM) (*p* < 0.05). After mixed silage, the content of DM of M1, M2 and M3 reached 449.70, 394.65 and 330.45 g·kg^−1^ DM, respectively, which were significantly higher than that of SY (*p* < 0.05). When rapeseed and *M. spicatum* were mixed at a 3:7 ratio (M6), the content of DM was 127.95 g·kg^−1^ FM higher than that in SZ, which was significantly higher than those in M4 and M5 (*p* < 0.05). 

It is apparent in Figure 2b that after 60 days of fermentation, the content of WSC of the mixed silage with rapeseed and alfalfa significantly increased (*p* < 0.05), and the content of WSC of M3 increased by 6.14% compared with that of the SX. The content of WSC of the rapeseed silage with 30% *M. spicatum* (M6) was 24.86 g·kg^−1^ DM, which was significantly higher than that of SZ (*p* < 0.05), but did not significantly differ from those of M1, M2 and M3 (*p* > 0.05).

Figure 2c shows that the content of CP in M1, M2 and M3 was 24.66, 30.97 and 21.07 g·kg^−1^ DM higher than that in SY, respectively. The content of CP reached 118.20 g·kg^−1^ DM when rapeseed and *M. spicatum* were mixed at 7:3 (M6), which was significantly higher than those of SY and SZ, and the rapeseed and alfalfa mixed silage groups (*p* < 0.05).

As shown in Figure 3a, the contents of NDF of M1 and M3 in the mixed rapeseed and alfalfa silage decreased to 47.63% and 51.80%. Compared with SY, the content of NDF of rapeseed silage with the addition of *M. spicatum* (M4, M5 and M6) decreased by 15.50%, 21.58% and 15.90%, respectively.

Figure 3b shows that the content of ADF in M1, M2 and M3 was 33.87%, 27.35% and 27.43%, respectively, they were significantly lower than those of SY and SZ (44.95% and 46.27%) (*p* < 0.05). The content of ADF after the silage of rapeseed and *M. spicatum* at 7:3 (M6) was 52.65%, which was 10.28% lower than that of SZ, but significantly higher than that of M2 (*p* < 0.05).

### 3.4. Analysis of the Fermentation Quality of Mixed Silage

The organic acid, pH and NH_3_-N/TN contents of the silage were determined after 60 days of fermentation, and the results are shown in Table 6. The silage materials (*p* < 0.01) had a significant effect on the content of DM in feed. The pH of YX decreased with the increase in the ratio of alfalfa added. When the mixing ratio was 3:7, the pH was 3.95, which was significantly lower than that in the 5:5 and 7:3 mixed silage groups (*p* < 0.05). The pH in YZ showed an upward trend with the increase in amount of *M. spicatum* added. The content of NH_3_-N/TN was affected by the silage materials (*p* < 0.01), mixing ratio (*p* < 0.01) and their interaction (*p* < 0.01). The content of NH_3_-N/TN in SY was the highest (7.89%). When mixed with alfalfa (YX) or *M. spicatum* (YZ), the content decreased to varying degrees. The content of NH_3_-N/TN of YZ was significantly lower than that in the YX (6.77%) (*p* < 0.05). The silage material (*p* < 0.01), mixing ratio (*p* = 0.029) and their interaction (*p* < 0.01) affected the content of LA. The content of LA of YX was significantly higher than that of YZ (67.64 and 26.29 g·kg^−1^ DM) (*p* < 0.05). The content of LA of YX increased with the increase in addition of alfalfa, and the content was the highest (96.46 g·kg^−1^ DM) at a 3:7 mixing ratio (*p* < 0.05). The content of BA in SX and SZ was 1.79 and 1.34 g·kg^−1^ DM, after silage with rape, respectively. The content of BA in YX and YZ decreased to some extent, and the latter was significantly lower than the former (1.47 and 0.99 g·kg^−1^ DM) (*p* <0.05).

### 3.5. Quality Evaluation of the V-Score in Different Treatment Groups

After the end of the silage period, the V-Score scoring system was used to evaluate the fermentation quality of the silage. As shown in Table 7, the silage quality score of each treatment group > 70, and the score of SZ in the single silage group was higher and reached 79.19. The mixed silage group of rapeseed and alfalfa showed a general performance. The scores of M2 and M3 were 75.43 and 73.66, respectively, and the score of M1 was higher than that of the others. The fermentation quality of the mixed silage group of rapeseed and *M. spicatum* had a higher fermentation quality. The scores of M5 and M6 reached 89.15 and 89.40, respectively, demonstrating good performance.

### 3.6. Comprehensive Evaluation of the Membership Function of Different Treatment Groups

The fuzzy membership function was used to calculate and analyze the nutrition and fermentation indices of the mixed silage of rapeseed and alfalfa or *M. spicatum*. A larger function value indicated a higher quality of the silage under the mixed ratio. As shown in Table 8, when the mixing ratio of rapeseed and alfalfa was 3:7, the function value reached its maximum (0.7068), which was 33.81% higher than that of M3 and 1.72 times that of SY. In the mixed silage group of rapeseed and *M. spicatum*, the trend was the opposite. The function value of M6 was the highest (0.7439), at 1.96- and 3.24-fold that of M4 and SZ, respectively. The subordinate function values of each group were ranked as follows: M6 > M1 > M5 > M2 > M3 > SX > M4 > SY > SZ.

### 3.7. Effect of Additives on the Nutritional Quality of Mixed Silage

As shown in Figure 4a, compared with U1, the DM content of U1 and M1 decreased by 39.90 and 29.75 g·kg^−1^, respectively. The content of DM of U2, M2 and MU2 were significantly lower than that of C2 (391.45 g·kg^−1^) (*p* > 0.05).

Figure 4b shows that after 60 days of fermentation, the content of WSC of UM1 was 18.67 g·kg^−1^ DM higher than that of C1 (25.21 g·kg^−1^ DM). The content of WSC in M2 was 34.45 g·kg^−1^ DM, which was significantly higher than those in C2 and U2 (24.78 and 25.03 g·kg^−1^ DM) (*p* < 0.05).

As shown in Figure 4c, the content of CP of MU1 was significantly higher than that of the other treatments (*p* < 0.05), which was 28.76 g·kg^−1^ DM higher than that of C1. The CP content of UM2 and U2 slightly increased compared with that of C1, but there was no significant difference between the two treatments (*p* > 0.05).

As shown in Figure 5a, the content of NDF of M1 and UM1 decreased by 10.45% and 10.88% compared with that in C1 (47.63%), respectively. The content in U2 was significantly higher than that of C2 (54.48% and 41.73%) (*p* < 0.05).

It is apparent in Figure 5b that after the silage, the content of ADF in the rapeseed and alfalfa additive treatment groups (U1, M1 and MU1) had no significant difference with that of C1. In the mixed silage group of rapeseed and M. spicatum, the content of ADF of UM2 and U2 had no significant difference to C1, while the content of ADF of M2 decreased by 9.15%.

### 3.8. Effect of Additives on the Fermentation Quality of Mixed Silage

After 60 days of silage, the influence of additives on the fermentation quality of silage is shown in Table 9. The pH value of M1 was significantly the lowest (3.55) (*p* < 0.05), while the pH values of U1 and UM1 were significantly higher than those of C1 (3.95) (*p* < 0.05). The contents of NH_3_-N/TN in U1 and UM1 were significantly higher than that of C1 (*p* < 0.05). The NH_3_-N/TN of U2 was 10.82%, which was 7.65% higher than that of C2. The content of LA in UM1 was the highest (145.95 g·kg^−1^ DM) (*p* < 0.05), which was 49.49 g·kg^−1^ DM higher than that of C1. The content of LA in M2 (63.21 g·kg^−1^ DM) was significantly higher than those of U2 and UM2. C1 had the highest content of BA (1.41 g·kg^−1^ DM) (*p* < 0.05), and U1, M1 and UM2 all had lower levels compared with that of C1. There was no significant difference in the content of BA between UM2 and C2 (*p* > 0.05), but both groups were significantly higher than that of M2 (0.46 g·kg^−1^ DM) (*p* < 0.05). The content of LA/AA in UM1 was 4.32, which was significantly higher than that in C1 (2.54) (*p* < 0.05). The LA/AA of U2 was 1.81, which was significantly higher than that of UM2 and C2 (*p* < 0.05).

## 4. Discussion

### 4.1. Chemical Composition of Materials

The moisture and sugar content of silage materials are important factors that affect the success of silage [27]. Zhang et al. [15] proposed that the silage raw materials should have a sufficient content of DM (approximately 350~500 g·kg^−1^ fresh matter [FM]). In this study, the content of DM of alfalfa was high (551. 63 g·kg^−1^ FM), while those of rapeseed and *M. spicatum* had a lower content of DM (274.03 and 281.27 g·kg^−1^ FM). To solve the problem of unsuitable material moisture, air drying before silage or mixed silage can be used to regulate the content of water in the materials [28]. Dunière et al. [29] concluded that the initial content of WSC should be more than 70 g·kg^−1^ DM, which facilitates the production of organic acids during the fermentation process. Alfalfa and *M. spicatum* did not have enough WSC (63.14 and 54.15 g·kg^−1^ DM), but the nutrient complementation and balance could be achieved after mixed silage with rapeseed (WSC: 114.48 g·kg^−1^ DM), which was similar to the research of Zhang et al. [30]. 

### 4.2. Effect of Mixed Silage of Rapeseed and Alfalfa or M. spicatum on Silage Quality

When rapeseed was mixed with alfalfa or M. spicatum, the content of dry matter (DM) in silage was effectively regulated. Rezende et al. [31] also proved that silage materials with high moisture content should not be silaged alone, and mixed silage with crops with a high content of DM can regulate the content of water. After 60 days, the content of WSC of the three silage materials decreased to differing degrees because WSC, as the primary fermentation substrate of silage, promoted the LA bacteria to produce LA from sugar in the feed during fermentation [32]. In this experiment, the content of WSC in rapeseed was relatively high. Adding rapeseed to alfalfa and *M. spicatum* can increase the content of WSC, which indicates that the mixed silage complements the carbohydrate, and reduces the consumption of WSC. This was similar to the results of Wang et al. [33] on the mixed silage of alfalfa, *Stylosanthus* and spicy wood leaves. CP is an important reference index to evaluate the nutritional quality of silage. A higher content indicates a higher nutritional value and vice versa [34]. After fermentation, the loss of content of CP in the mixed group of rapeseed and alfalfa increased with the increase in proportion of rapeseed, which could be because the fermentation environment with a high content of rapeseed facilitated the survival and reproduction of undesirable bacteria, resulting more protein degradation during fermentation. This was similar to the results of Li et al. [35]. The addition of a certain amount of alfalfa and *M. spicatum* can increase the content of CP in the silage. Similar to this study, the content of CP in forage increased when leguminous forage was silaged with corn and sorghum (*Sorghum bicolor* L.) [36]. Kung et al. suggested that the formation of ammonia nitrogen was related to the degradation of CP, and NH_3_-N/TN in the silage revealed the degree of degradation of the CP in silage. A higher ratio indicates that more protein is degraded [37]. The content of NH_3_-N/TN of the high-quality silage should be lower than 10% [38]. In this study, the content of NH_3_-N/TN in each treatment group was lower than 10%, indicating that there was extensive protein hydrolysis. In addition, Valente et al. [39] concluded that the concentration of NH_3_-N was consistent with the degree of BA fermentation. The hydrolysis of more protein indicated a concentration of higher butyric acid, demonstrating that the quality of silage for fermentation was poor. The content of BA in M5 and M6 was significantly the lowest, which was consistent with the lower NH_3_-N/TN of the two groups. This showed that 30% and 50% of *M. spicatum* added to rapeseed silage could improve the fermentation quality of the feed. The pH is the simplest and most direct index to measure the fermentation quality of silage. It is generally believed that the pH of high-quality silage is between 3.8 and 4.2 [40], and the decrease in pH in the silage is negatively related to the accumulation of LA [38]. In this study, the high pH (5.66) of SZ indicated that silage alone is not suitable to preserve the *M. spicatum* feed, which could be owing to the insufficient fermentation substrate of *M. spicatum* (a low content of WSC), resulting in the low production of lactic acid. The pH increased with the increase in the proportion of *M. spicatum*, which could occur because relatively high amounts of rapeseed provide sufficient fermentation substrate for the silage process. Gallego et al. found similar results when they used *Elodea* and wheat straw for mixed silage [13]. When rapeseed and alfalfa were mixed at a ratio of 3:7 for 60 days, the pH decreased to 3.95, and the content of LA reached 96.46 g·kg^−1^ DM. This could be because the content of WSC in the mixed silage was optimally complementary at this ratio, which provided sufficient fermentation substrate for the growth and reproduction of LA bacteria, thereby increasing the content of LA in the silage and ultimately reducing the pH. This result was consistent with the findings of Jiménez-Calderón et al. [41]. 

ADF and NDF are used as conventional indices to evaluate the quality of fiber. Falkner et al. [42] concluded that the content of NDF would affect the palatability of feed and the rate of animal feeding. A study by Kaplan [43] showed that the contents of ADF and NDF in the alfalfa and sorghum mixed silage were lower than that in a single silage group. In this study, the content of NDF in M1 of the rapeseed and alfalfa mixed silage group was relatively low, which could be because the concentrations of structural carbohydrates in the cell wall of alfalfa was lower than that of rape, and the mixed silage improved the digestibility of silage [44]. Compared with the mixed silage group, the higher ADF content in the single silage group could be related to the increase in temperature during microbial respiration [45], which leads to the combination of reducing sugars and amino compounds, a process called the “Maillard reaction.” A higher temperature is an important factor that promotes the Maillard reaction, thus affecting the ADF content and ultimately damaging the nutritional value of the feed [46].

### 4.3. Effect of Additives on the Quality of Mixed Silage of Rapeseed and Alfalfa or M. spicatum

Chen et al. [47] concluded that the addition of molasses could improve the sugar required for the growth of LA bacteria during the fermentation process, provide more fermentation substrate and make the reaction closer to a homolactic acid fermentation. Researchers found that adding molasses could increase the content of dry matter, lactic acid and water-soluble carbohydrate in silage, reduce the pH, inhibit butyric acid fermentation and improve the fermentation quality of silage [48,49]. In this experiment, compared with the control group (C1 and C2), the content of LA of the groups added with 3% molasses alone (M1 and M2) increased, and LA/AA was high, which indicated that homolactic acid fermentation was dominant during silage after adding molasses [49]. However, the contents of NH_3_-N/TN of M2 and M1 were slightly higher than those of the control group, which could be related to the decomposition of a small amount of crude protein (mostly nonprotein nitrogen), which is contained in molasses [38]. Simultaneously, we found that the contents of DM and CP in M1 and M2 had no significant change, or even slightly decreased, compared with the control group. This could be owing to some adverse effects of molasses, such as the increase in exudate loss during silage, particularly the loss of nutrient quality under high moisture conditions [50]. Arbabi et al. [48] concluded that molasses could promote the degradation of structural carbohydrates. The content of NDF in the molasses addition group (M1 and M2) in this test was significantly lower than that in the control group, which was consistent with this viewpoint.

Urea is a nutritional additive that is commonly used in silage technology, which can increase the crude protein of silage. Santos et al. found that the appropriate amount of urea (0.1756~1.4048%) positively correlated with the content of CP in sorghum silage fermentation materials [51]. In this experiment, the content of CP in the groups added with 0.3% urea alone (U1 and U2) was significantly higher than that of the control groups (C1 and C2), which was consistent with the viewpoint described above. However, we found that U1 and U2 lost DM, and there were higher levels of pH and NH_3_-N/TN, which led to poor fermentation quality; this was consistent with the research of Yunus et al. [6]. This is because the OH^–^ generated from dissolution of urea combines with H^+^ in the environmental medium, which hinders the decrease in pH and ultimately inhibits the activity of lactic acid bacteria, leading to the decrease in content of LA [52]. Alternatively, Cesareo et al. [53] found that urease can promote the decomposition of urea in a higher pH environment. In this study, the higher pH of U1 and U2 provided conditions for the decomposition of urea. This result was consistent with the result of higher NH_3_-N/TN in U1 and U2 in Table 9. Urea also affects the fiber quality of silage. Wanapat and Phesatcha found that adding urea can reduce the contents of ADF and NDF and improve the quality of silage [54,55]. However, in this experiment, the contents of ADF and NDF in U2 were higher than those in C2, which could be owing to the enhanced activity of harmful microorganisms in the high pH (5.56) environment, which results in the large decomposition of nutrients (primarily soluble carbohydrate) in silage, and finally resulting in a relatively high content of structural carbohydrates. However, the mechanism of the role of urea in the degradation of NDF and ADF has not been confirmed.

In this study, 3% molasses and 0.3% urea compound additive groups (UM1 and UM2) were simultaneously established. The pH, NH_3_-N/TN and content of BA of the rapeseed and alfalfa compound additive group (UM1) were lower than those of U1, indicating that adding molasses was an effective measure to counteract the adverse effects of urea. It could alleviate the increase in pH and inhibit the decomposition of urea by urease. Compared with U2, the pH and contents of NH_3_-N/TN and BA of UM2 were decreased, but were significantly higher than those of M2, and the lactic acid content: M2 > UM2 > U2, which indicated that adding 3% molasses alone could effectively improve the fermentation quality of mixed silage of rapeseed and *M. spicatum*. However, the mechanism and potential role of the mixed addition of molasses and urea in improving the quality of silage merits further study.

## 5. Conclusions

In conclusion, the V-Score and membership function value were higher when rapeseed and alfalfa or *M. spicatum* were silaged at 3:7 and 7:3, respectively. Simultaneously, adding a compound additive (3% molasses + 0.3% urea) to the mixed silage of rapeseed and alfalfa (3:7) and adding 3% molasses to the mixed silage of rapeseed and *M. spicatum* (7:3) can more effectively improve the nutrition and fermentation quality of mixed silage.

This study provides a promising method to make the three biomass raw materials fully converted and utilized, reducing agricultural resource waste.

## Figures and Tables

**Figure 1 ijerph-20-03884-f001:**
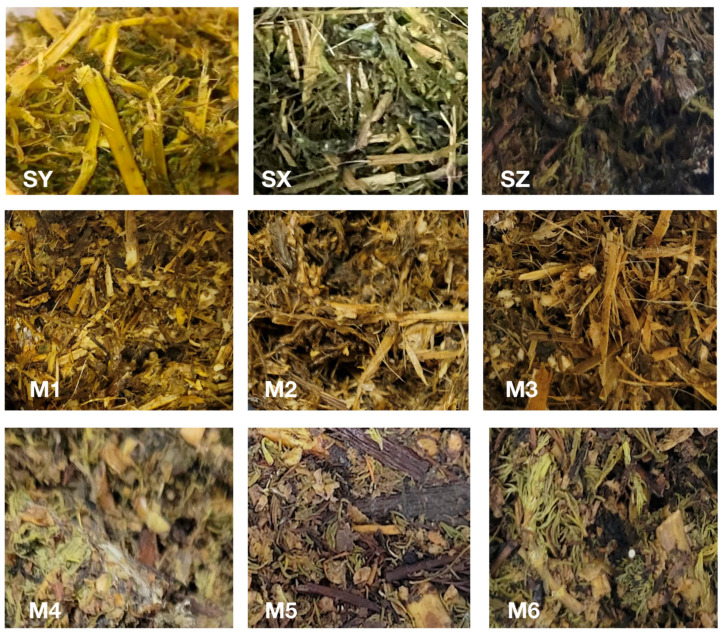
The sensory changes of mixed silage after the cans were opened. Note: SY-Rapeseed silage; SX-Alfalfa silage; SZ-*M. spicatum* silage; M1-Rapeseed × Alfalfa (3:7); M2-Rapeseed × Alfalfa (5:5); M3- Rapeseed × Alfalfa (7:3); M4-Rapeseed × *M. spicatum* (3:7); M5-Rapeseed × *M. spicatum* (3:7); M6-Rapeseed × *M. spicatum* (7:3). The same below.

**Figure 2 ijerph-20-03884-f002:**
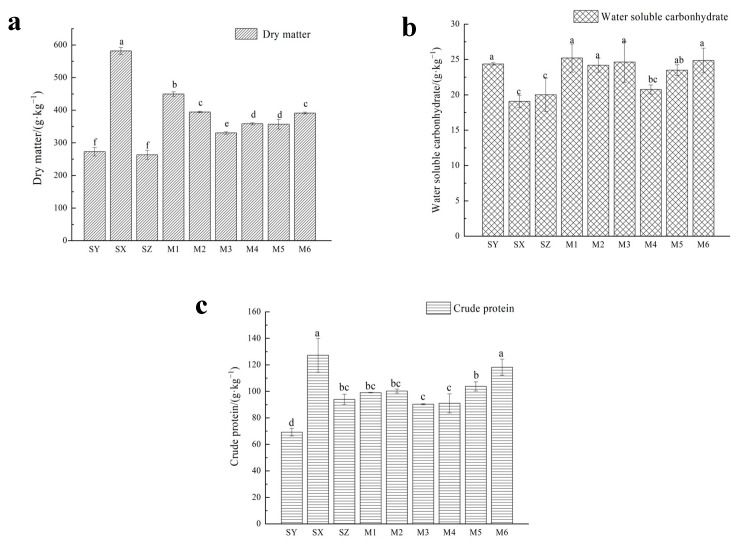
Contents of DM (**a**), WSC (**b**) and CP (**c**) after mixed silage of rapeseed with alfalfa or *Myriophyllum spicatum* in different proportions. Note: Different lowercase letters indicate significant difference between different treatment groups (*p* < 0.05). The same below.

**Figure 3 ijerph-20-03884-f003:**
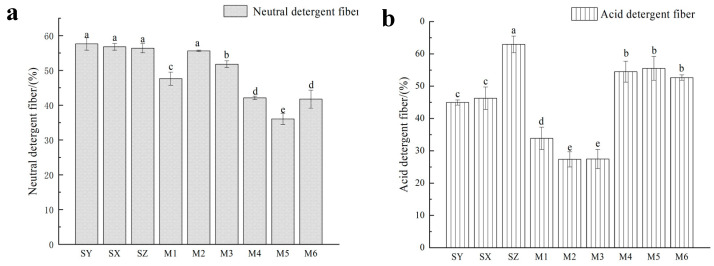
Contents of NDF (**a**) and ADF (**b**) after mixed silage of rapeseed with alfalfa or *Myriophyllum spicatum* in different proportions. Note: Different lowercase letters indicate significant difference between different treatment groups (*p* < 0.05).

**Figure 4 ijerph-20-03884-f004:**
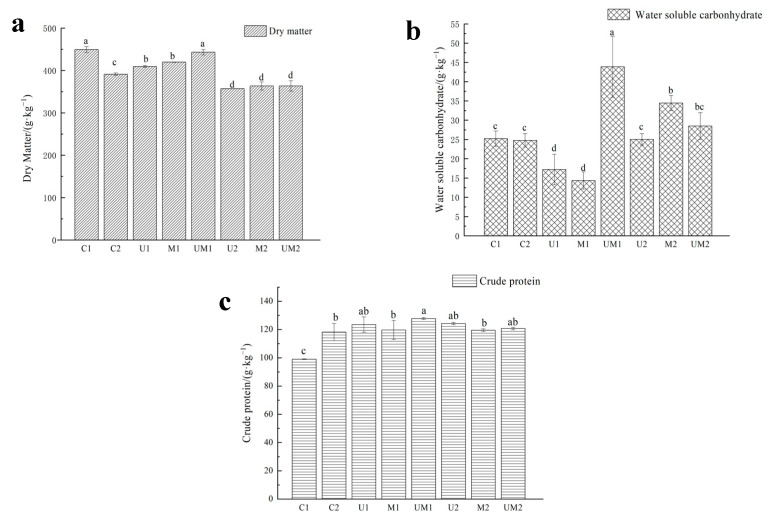
Effects of additives on DM (**a**), WSC (**b**) and CP (**c**) content in mixed silage. Note: C1-Rape × Alfalfa (3:7); C2-Rapeseed×*M. spicatum* (7:3); U1, M1 and MU1 indicate that 0.3% urea, 3% molasses, 0.3% urea and 3% molasses (all fresh weight) are successively added when rape and alfalfa (3:7) are mixed for silage. U2, M2 and MU2 indicate that 0.3% urea, 3% molasses, 0.3% urea and 3% molasses (all fresh weight) are successively added during the mixed silage of rapeseed and *M. spicatum* (7:3). The same below. Note: Different lowercase letters indicate significant difference between different treatment groups (*p* < 0.05).

**Figure 5 ijerph-20-03884-f005:**
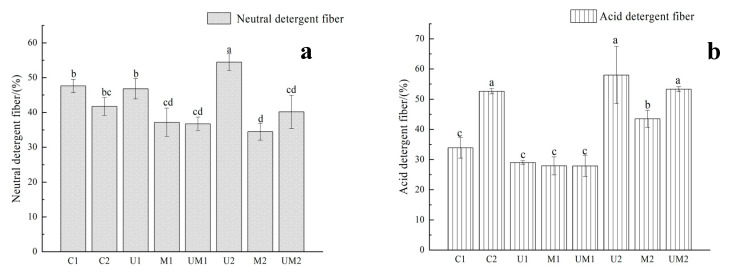
Effect of additives on the contents of NDF (**a**) and ADF (**b**) in mixed silage. Note: Different lowercase letters indicate significant difference between different treatment groups (*p* < 0.05).

**Table 1 ijerph-20-03884-t001:** Chemical composition of silage raw materials.

Ensilage Materials	Dry Matter (g·kg^−1^ FM)	Crude Protein (g·kg^−1^ DM)	Water Soluble Carbohydrate (g·kg^−1^ DM)	Acid Detergent Fiber (% DM)	Neutral Detergent Fiber (% DM)
Rapeseed	274.03 ± 1.21 B	51.15 ± 0.82 C	114.48 ± 1.70 A	33.07 ± 1.04 B	47.34 ± 1.21 B
Alfalfa	551.63 ± 4.12 A	186.76 ± 3.43 A	63.14 ± 3.04 B	35.11 ± 0.81 B	46.10 ± 0.72 B
*M. spicatum.*	281.27 ± 4.57 B	149.70 ± 0.59 B	54.15 ± 1.59 B	48.32 ± 0.95 A	53.22 ± 0.54 A

Note: Different capital letters indicate significant difference between different treatment groups (*p* < 0.05).

**Table 2 ijerph-20-03884-t002:** Treatments of test 1.

Material	Treatment								
	SY	SX	SZ	M1	M2	M3	M4	M5	M6
Rapeseed	1	-	-	3	5	7	3	5	7
Alfalfa	-	1	-	7	5	3	-	-	-
*M. spicatum.*	-	-	1	-	-	-	7	5	3

**Table 3 ijerph-20-03884-t003:** Treatments of test 2.

Treatment	Mixture	Additive
C1	YX37	None
U1		0.3% urea
M1		3% molasses
UM1		0.3 urea + 3% molasses
C2	YZ73	None
U2		0.3% urea
M2		3% molasses
UM2		0.3 urea + 3% molasses

**Table 4 ijerph-20-03884-t004:** Abbreviations.

Symbol	Description
DM	Dry matter
FM	Fresh matter
WSC	Water soluble carbohydrate
CP	Crude protein
NDF	Neutral detergent fiber
ADF	Acid detergent fiber
SY	Rapeseed silage
SX	Alfalfa silage
SZ	*Myriophyllum spicatum* silage
YX	Mixed silage group of rapeseed and alfalfa
YZ	Mixed silage group of rapeseed and *Myriophyllum spicatum*
LA	Lactic acid
AA	Acetic acid
PA	Propionic acid
BA	Butyric acid
NH_3_-N	Ammonia nitrogen
TN	Total nitrogen
SEM	Standard error of mean

**Table 5 ijerph-20-03884-t005:** Sensory evaluation of mixed silage after the cans were opened.

Group	Mildew	Smell		Color		Quality of Material		Composite Scores	
		Describe	Average Score	Describe	Average Score	Describe	Average Score	Total Score	Grade
SY	A little mildew	Strong sour smell, weak aromatic smell	6.00	Bright yellow, Slightly white	1.00	The stem and leaf structure are well protected, hard and slightly sticky	2.33	9.33	Level III
SX	A little mildew at the bottle mouth	Strong ammonia taste, almost no acid taste	6.00	green and slightly white	1.00	The stem and leaf structure are well protected, hard and loose	1.33	8.33	Level III
SZ	A little mildew	A faint sour smell	7.00	Dark brown, slightly white	0.67	The stem and leaf structure are well protected and loose	2.00	9.67	Level III
M1	No mildew	No sour odor, aromatic fruit flavor	11.33	Yellow and slightly green	1.67	The stem and leaf structure is well protected, soft and loose.	3.33	15.00	Level II
M2	No mildew	No sour odor, aromatic fruit flavor	11.33	Yellow, slightly green	1.33	The stem and leaf structure is well protected, soft and loose.	2.67	14.00	Level II
M3	No mildew	A faint smell of bread	9.33	Slightly white and medium yellow	1.33	The stem and leaf structure is well protected, soft and loose.	2.67	14.00	Level II
M4	A little mildew	Weak aromatic taste	8.33	Brown and slightly yellow	1.00	The stems and leaves are not rotten, and the texture is soft and loose	2.67	12.00	Level II
M5	No mildew	No sour smell, weak aromatic fruit flavor	9.33	Yellow brown, slightly green	1.33	The stem and leaf structure is well protected, soft and loose.	3.00	13.67	Level II
M6	No mildew	No sour odor, aromatic fruit flavor	10.33	Light yellowish brown, slightly green	1.3	The stem and leaf structure is well protected, soft and loose.	3.33	15.00	Level II

**Table 6 ijerph-20-03884-t006:** Fermentation quality of the mixed silage of rapeseed with alfalfa or *Myriophyllum spicatum* in different proportions.

Item	SY	SX	SZ	Ratio	YX	YZ	Mean	SEM	*p*		
									Species	Ratio	S × R
pH	4.71	4.47	5.66	3:7	3.95 ^Bc^	5.32 ^Aa^	4.63	0.031	<0.001	0.154	<0.001
				5:5	4.13 ^Bb^	5.08 ^Ab^	4.60				
				7:3	4.77 ^Ba^	4.56 ^Ac^	4.66				
Mean					4.40 ^B^	5.06 ^A^					
NH_3_-N/TN(%)	7.89	7.33	5.07	3:7	4.61 ^Ac^	5.19 ^Aa^	4.90 ^b^	0.220	<0.001	<0.001	<0.001
				5:5	5.69 ^Ab^	3.32 ^Bb^	4.51 ^b^				
				7:3	8.35 ^Aa^	3.17 ^Bb^	5.76 ^a^				
Mean					6.77 ^A^	4.32 ^B^					
LA/ (g·kg^−1^ DM)	34.09	63.74	12.93	3:7	96.46 ^Aa^	18.88 ^Bc^	57.67 ^a^	2.245	<0.001	0.029	<0.001
				5:5	81.98 ^Ab^	25.24 ^Bb^	53.61 ^b^				
				7:3	61.95 ^Ac^	40.29 ^Ba^	51.12 ^b^				
Mean					67.64 ^A^	26.29 ^B^					
AA/ (g·kg^−1^ DM)	16.03	40.55	75.75	3:7	38.72 ^A^	16.90 ^B^	27.81	3.791	<0.001	0.622	0.817
				5:5	41.37 ^A^	18.16 ^B^	29.76				
				7:3	35.27 ^A^	16.78 ^B^	26.02				
Mean					34.39 ^A^	28.72 ^A^					
PA/ (g·kg^−1^ DM)	6.31	3.95	5.27	3:7	7.24 ^B^	17.23 ^Aa^	12.24 ^a^	0.771	<0.001	<0.001	<0.001
				5:5	7.21 ^B^	12.17 ^Ab^	9.69 ^b^				
				7:3	7.18 ^A^	5.48 ^Bc^	6.33 ^c^				
Mean					6.38 ^B^	9.29 ^A^					
BA/ (g·kg^−1^ DM)	1.28	1.79	1.34	3:7	1.41 ^Ab^	1.29 ^Aa^	1.35 ^a^	0.084	<0.001	<0.001	<0.001
				5:5	1.65 ^Aa^	0.53 ^Bb^	1.09 ^b^				
				7:3	1.21 ^Ab^	0.53 ^Bb^	0.87 ^c^				
Mean					1.47 ^A^	0.99 ^B^					
LA/AA	2.13	1.57	0.17	3:7	2.54 ^Aa^	1.12 ^Bc^	0.95 ^b^	<0.01	<0.001	0.057	<0.001
				5:5	1.98 ^Aab^	1.39 ^Bb^	1.13 ^a^				
				7:3	1.76 ^Bb^	2.40 ^Aa^	1.02 ^ab^				
Mean					2.00 ^A^	1.44 ^B^					

Note: Values followed by the same lowercase letters indicate no significant difference across different mixing ratios in the same grass (*p* < 0.05, vertical comparison); Values followed by the same capital letters indicate no significant difference across different grass in the same mixing ratios (*p* < 0.05, horizontal comparison).

**Table 7 ijerph-20-03884-t007:** Quality evaluation of the V-Score in different treatment groups.

Item	Treatment	(NH_3_-N/TN)		AA + PA		BA		Scores	Grade
		Ratio	Scores	Content/%DM	Scores	Content/%DM	Scores		
V-Score	SY	7.89	44.22	2.23	0.00	0.13	29.73	73.96	common
	SX	7.33	45.35	4.45	0.00	0.18	25.67	71.02	common
	SZ	5.03	49.93	8.10	0.00	0.13	29.26	79.19	common
	MI	4.61	50.78	4.60	0.00	0.14	28.76	79.54	common
	M2	5.69	48.61	4.86	0.00	0.16	26.82	75.43	common
	M3	8.35	43.41	4.25	0.00	0.12	30.36	73.66	common
	M4	5.19	49.63	3.41	0.00	0.13	29.71	79.34	common
	M5	3.32	53.35	3.03	0.00	0.05	35.80	89.15	good
	M6	3.17	53.67	2.23	0.00	0.05	35.73	89.40	good

**Table 8 ijerph-20-03884-t008:** Membership function value and comprehensive ranking.

Item	Membership Function Value								
	SY	SX	SZ	M1	M2	M3	M4	M5	M6
DM/(g/kg)	0.0297	1.0000	0.0000	0.5849	0.4119	0.2417	0.2981	0.2945	0.5338
CP/(g·kg^−1^ DM)	0.0000	1.0000	0.4256	0.5136	0.5345	0.3637	0.3745	0.5958	0.8441
WSC/(g·kg^−1^ DM)	0.8627	0.0000	0.1535	1.0000	0.8352	0.9080	0.2746	0.7222	0.9435
NDF/(% DM)	0.0000	0.0394	0.0579	0.4635	0.0939	0.2700	0.7184	1.0000	0.7370
ADF/(% DM)	0.5053	0.4683	0.0000	0.8168	1.0000	0.9977	0.2375	0.2087	0.2888
pH	0.5543	0.6979	0.0000	1.0000	0.8974	0.5220	0.1994	0.3402	0.6422
NH_3_-N/TN/(%)	0.0884	0.1377	0.6304	0.7194	0.5106	0.0000	0.6109	0.9679	1.0000
LA/(g·kg^−1^ DM)	0.2533	0.6083	0.0000	1.0000	0.8266	0.5868	0.0713	0.1474	0.3275
AA/(g·kg^−1^ DM)	1.0000	0.5894	0.0000	0.6201	0.5757	0.6778	0.9854	0.9644	0.9875
PA/(g·kg^−1^ DM)	0.8221	1.0000	0.9011	0.7522	0.7547	0.7572	0.0000	0.3809	0.8853
BA/(g·kg^−1^ DM)	0.4007	0.0000	0.3543	0.3047	0.1134	0.4735	0.3985	1.0000	0.9929
Mean	0.4106	0.5037	0.2294	0.7068	0.5958	0.5282	0.3790	0.6020	0.7439
Ranking	8	6	9	2	4	5	7	3	1

**Table 9 ijerph-20-03884-t009:** Effect of additives on the fermentation quality of mixed silage.

Item	Treatments									
	C1	C2	U1	M1	UM1	U2	M2	UM2	SEM	*p*-Value
pH	3.95 ^e^	4.56 ^d^	4.72 ^bc^	3.55 ^f^	4.60 ^cd^	5.56 ^a^	4.73 ^b^	4.70 ^bc^	0.032	<0.001
NH_3_-N/TN (%)	4.61 ^d^	3.17 ^e^	9.25 ^b^	5.05 ^d^	8.00 ^c^	10.82 ^a^	4.66 ^d^	9.68 ^b^	0.194	<0.001
LA//(g·kg^−1^ DM)	96.46 ^c^	40.29 ^f^	56.90 ^de^	133.71 ^b^	145.95 ^a^	55.752 ^e^	63.21 ^d^	53.06 ^e^	2.050	<0.001
AA//(g·kg^−1^ DM)	38.72 ^bc^	16.78 ^f^	46.11 ^a^	41.28 ^ab^	33.77 ^cd^	30.97 ^d^	14.72 ^f^	24.18 ^e^	1.800	<0.001
PA//(g·kg^−1^ DM)	7.24 ^a^	5.48 ^b^	2.13 ^d^	1.63 ^e^	1.93 ^de^	2.74 ^c^	2.34 ^cd^	2.65 ^c^	0.139	<0.001
BA//(g·kg^−1^ DM)	1.41 ^a^	0.53 ^cd^	0.82 ^b^	0.58 ^cd^	0.64 ^c^	0.66 ^bc^	0.46 ^d^	0.55 ^cd^	0.047	<0.001
LA/AA	2.54 ^c^	2.40 ^cd^	1.24 ^e^	3.26 ^b^	4.32 ^a^	1.81 ^de^	4.62 ^a^	2.23 ^cd^	0.177	<0.001

Note: Data are the means of three samples. Values followed by the same lowercase letters indicate no significant difference between treatments (*p* < 0.05).

## Data Availability

All data generated or analysed during the study are included in this published article. The datasets used and/or analysed during the present study are available from the corresponding author upon reasonable request.

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
