# Peer review of "Study on the Quality of Mixed Silage of Rapeseed with Alfalfa or Myriophyllum"

_ijerph, 2023, doi:10.3390/ijerph20053884_

Round 1
Reviewer 1 Report
The manuscript entitled “Study on the Quality of Mixed Silage of Rapeseed with Alfalfa or Myriophyllum spicatum” is very interesting for the scientific community. My comments to the authors are listed below.
Title: I think that the title of the manuscript should be more general and avoid the combined use of non-scientific names for some plants (i.e., rapeseed, alfalfa) and the scientific name for another plant (i.e., M. spicatum).
Abstract: Please write in italic the name of the genus and species in the abstract.
Keywords: Please give only the relevant keywords for the present work.
Introduction: Please write in a more concise way the introduction of the manuscript by kipping only the literature data that are relevant to the current work. There are too many details that make it difficult to follow this section.
Please put a paragraph when you start to tock about a new plant type (i.e., before alfalfa, before M. spicatum) in the introduction section.
Materials and methods, Results, Discussion: The manuscript is well written, although there are too many abbreviations that make it difficult to follow what the authors wanted to say. Will be easier for the readers if you avoid, as much as you can the abbreviation in the manuscript. Please present yours results in more synthetic way.
Why did the authors choose the concentration of 0.3% for urea and 3% for molasses?
Please write in italic the name of the genus and species in the legend of figures and tables.
Conclusions: Please write more concisely and emphasize the importance of the present work.
Author Response
Title: I think that the title of the manuscript should be more general and avoid the combined use of non-scientific names for some plants (i.e., rapeseed, alfalfa) and the scientific name for another plant (i.e., M. spicatum).
I have noticed this problem and have completed relevant modifications in the manuscript. I decided to use non-scientific names uniformly.
Abstract: Please write in italic the name of the genus and species in the abstract.
I have completed relevant modifications in the manuscript.
Keywords: Please give only the relevant keywords for the present work.
Partial deletion has been made in this part.
Introduction: Please write in a more concise way the introduction of the manuscript by kipping only the literature data that are relevant to the current work. There are too many details that make it difficult to follow this section.
Please put a paragraph when you start to tock about a new plant type (i.e., before alfalfa, before M. spicatum) in the introduction section.
I have completed relevant modifications in the manuscript.
Materials and methods, Results, Discussion: The manuscript is well written, although there are too many abbreviations that make it difficult to follow what the authors wanted to say. Will be easier for the readers if you avoid, as much as you can the abbreviation in the manuscript. Please present yours results in more synthetic way.
Table 4 (abbreviation) has been added to section 2.4.
Why did the authors choose the concentration of 0.3% for urea and 3% for molasses?
Throug literature review I learned that the dosage of silage additives is often different when different silage materials are mixed. For example, molasses additive(3%) had improved the fermentation quality and tastes of alfalfa silage,this group obtained the ideal pH value (below 4.5) and the best condition for long-term preservation.The addition of 0.5% urea plus 8% molasses improved DM, CP, NDF and ADF content of canola silages better than other additive levels of urea and molasses. In this experiment, I referred to these data and determined the approximate dose range, so I finally chose 3% molasses and 0.3% urea to explore their influence on silage quality.
Please write in italic the name of the genus and species in the legend of figures and tables.
I have completed relevant modifications in the manuscript.
Conclusions: Please write more concisely and emphasize the importance of the present work.
I have revised and improved the conclusions.

Reviewer 2 Report
I recommend revision of the manuscript. Please find detailed comments below:
1. Introduction part needs editing. There are too many repetitions here, e.g. "not only helps to, but also helps to" (lines 42-44), "the feed source (...) value of feed" (44-45), "additive, addition, additives, additives" (lines 49- 50).
2. Additionally, please edit sentences that are too long, such as in lines 66-69 or 174-175. I recommend using the help of a native speaker to make significant text edits of vocabulary and style throughout the whole manuscript.
3. Please add reference to line 33-34 - about the production of agricultural waste per year.
4. Please organize the introduction part and divide the description into clear paragraphs about each of the tested plants - specify what problems lie at the bottom of this research, why were these mixtures made? This is confusing for the reader at this point.
5. Line 104 - the expertise provides, not: should provide..
6. Materials and methods - line 116: how long did you naturally dry the materials?
7. Line 118 - please indicate what chemical components of the material were analyzed (e.g. indicate the next section that discusses it so that the reader does not look for it).
8. Please explain on what basis the ratios of substrates used were selected.
9. It would be helpful to show the experiment matrix in the form of table/diagram with all treatments (lines 134-139).
10. Results: The manuscript is overwhelmed with descriptions of results; it is necessary to shorten them and to indicate the main discovered trends, e.g. which variants were the most/least favourable. This applies to both sections 3.1, 3.7 and 3.8. Please do not discuss each of the values ​​that are also presented in the tables (duplication of results, e.g. section 3.1).
11. Results: I recommend editing the text again, in lines 179-192 there are many repetitions of words such as "score", "color", "odor", often in the same sentence.
12. Figure 1 - please consider changing the variant markings from red to a more visible one.
13. Section 3.2 - it is necessary to explain in the text/tables the abbreviations DM, FM, CP, ADF, DNF etc.
14. In addition, the accumulation of many abbreviations in one sentence makes it difficult for the reader to focus on the research data (among others, section 3.4).
15. Please consider changing the order of sections 3.1 and 3.2.
16. Figures 2, 3, 4, 5 - captions of subsequent parts of figures (a, b, c) should be in figures captions, not ON figures.
17. Discussion: conclusions in lines 441-444, 446-447 and 452-454 are not supported by the performed tests (no microbiological analyses). Therefore, there can be no certainty about these premises.
18. Lines 516-519 - repetition of information from the results section.
19. Lines 553-556 - this information indicates that the appropriate doses of urea have already been developed. Therefore, please indicate what new results the conducted research brought.
20. Conclusions: lines 600-601 - please edit, research has not proven that the problem of agricultural resource waste has been solved.
Author Response
- Introduction part needs editing. There are too many repetitions here, e.g. "not only helps to, but also helps to" (lines 42-44), "the feed source (...) value of feed" (44-45), "additive, addition, additives, additives" (lines 49- 50).
I have revised and improved the Introduction.
- Additionally, please edit sentences that are too long, such as in lines 66-69 or 174-175. I recommend using the help of a native speaker to make significant text edits of vocabulary and style throughout the whole manuscript.
I have completed relevant modifications in the manuscript.
- Please add reference to line 33-34 - about the production of agricultural waste per year.
I have added the corresponding reference.
- Please organize the introduction part and divide the description into clear paragraphs about each of the tested plants - specify what problems lie at the bottom of this research, why were these mixtures made? This is confusing for the reader at this point.
I have revised and improved the Introduction.
- Line 104 - the expertise provides, not: should provide..
I have finished the modification.
- Materials and methods - line 116: how long did you naturally dry the materials?
This experiment silage materials were dried for 1 ~ 2 days, which has been indicated in the manuscript.
- Line 118 - please indicate what chemical components of the material were analyzed (e.g. indicate the next section that discusses it so that the reader does not look for it).
Table 1 (abbreviation) has been added to section 2.1.
- Please explain on what basis the ratios of substrates used were selected.
This experiment is a preliminary exploration of the suitable proportion of mixed silage. The two materials are mixed silage according to the fresh weight ratio of 3:7, 5:5 and 7:3 respectively, in order to find a reasonable mixing ratio. It is planned to carry out detailed research on other related ratios in the future.
- It would be helpful to show the experiment matrix in the form of table/diagram with all treatments (lines 134-139).
Table 2 and Table 3 has been added to section 2.2.
- Results: The manuscript is overwhelmed with descriptions of results; it is necessary to shorten them and to indicate the main discovered trends, e.g. which variants were the most/least favourable. This applies to both sections 3.1, 3.7 and 3.8. Please do not discuss each of the values that are also presented in the tables (duplication of results, e.g. section 3.1).
I have completed relevant modifications in the manuscript.
- Results: I recommend editing the text again, in lines 179-192 there are many repetitions of words such as "score", "color", "odor", often in the same sentence.
The text description in article 3.2 has been deleted, because I think the effect of treatment group (including color, odor, color, texture, etc.) can be directly seen in Table 5.
- Figure 1 - please consider changing the variant markings from red to a more visible one.
I have completed relevant modifications in the manuscript.
- Section 3.2 - it is necessary to explain in the text/tables the abbreviations DM, FM, CP, ADF, DNF etc.
Table 4 (abbreviation) has been added to section 2.4.
- In addition, the accumulation of many abbreviations in one sentence makes it difficult for the reader to focus on the research data (among others, section 3.4).
Table 4 (abbreviation) has been added to section 2.4.
- Please consider changing the order of sections 3.1 and 3.2.
I have changed the order of sections 3.1 and 3.2.
- Figures 2, 3, 4, 5 - captions of subsequent parts of figures (a, b, c) should be in figures captions, not ON figures.
I have completed relevant modifications in the manuscript.
- Discussion: conclusions in lines 441-444, 446-447 and 452-454 are not supported by the performed tests (no microbiological analyses). Therefore, there can be no certainty about these premises.
I have completed relevant modifications in the manuscript.
18.Lines 516-519 - repetition of information from the results section.
I have deleted that part.
- Lines 553-556 - this information indicates that the appropriate doses of urea have already been developed. Therefore, please indicate what new results the conducted research brought.
Relevant modifications have been completed in the manuscript.
New findings: Adding a compound additive (3% molasses +0.3% urea) to the mixed silage of rapeseed and alfalfa (3:7) and adding 3% molasses to the mixed silage of rapeseed and M. spicatum (7:3) can more effectively improve the nutrition and fermentation quality of mixed silage.
It has been explained in the third paragraph of section 3.3.
- Conclusions: lines 600-601 - please edit, research has not proven that the problem of agricultural resource waste has been solved.
I have revised and improved the conclusions.
